# JANUS-LoRA: A Balanced Low-Rank Adaptation for Continual Learning

**Cheng Chen** [1]  **Pengpeng Zeng** [2]  **Yuyu Guo** [3]  **Lianli Gao** [1]  **Hengtao Shen** [2]  **Jingkuan Song** [2 4]

## Abstract

Low-Rank Adaptation (LoRA) has emerged as a promising paradigm for Continual Learning. It independently updates its low-rank factors ($A$ and $B$), creating a composite update to the full weight matrix through their interaction. To prevent catastrophic forgetting, this update should remain orthogonal to the task-specific subspace that contains previously learned knowledge. However, we identify that this composite update systematically violates this orthogonality, reintroducing interference and undermining stability. Furthermore, naively enforcing this orthogonality compromises plasticity, disrupting the delicate stability-plasticity trade-off. To resolve these issues, we propose **Janus-LoRA**, a framework that restores this balance through two novel components. Specifically, we first introduce Gradient Rectification, a closed-form solution that mathematically decouples LoRA's factor updates, enforcing orthogonality against the historical knowledge subspace identified by an efficient Online Estimation. Next, to enhance plasticity, we introduce a Decoupled Margin Loss that promotes feature-level separation by pushing new feature representations away from old ones, thus creating distinct, low-interference regions for new learning. Comprehensive experiments on challenging benchmarks demonstrate that by harmonizing parameter-level orthogonality with feature-level separation, Janus-LoRA achieves a superior balance and establishes new state-of-the-art performance. Codes are publicly available at https://github.com/zackschen/Janus-LoRA.

[1]School of Computer Science and Engineering, University of Electronic Science and Technology of China, Chengdu, China [2]School of Computer Science and Technology, Tongji University, Shanghai, China [3]Independent Researcher [4]Shanghai Innovation Institute, Shanghai, China. Correspondence to: Jingkuan Song <jingkuan.song@gmail.com>.

*Proceedings of the 43rd International Conference on Machine Learning*, Seoul, South Korea. PMLR 306, 2026. Copyright 2026 by the author(s).

## 1. Introduction

Continual Learning (CL) (Schwarz et al., 2018; Ayub & Wagner, 2021; Chen et al., 2022), the capacity of an artificial agent to learn incrementally from a sequence of tasks, is considered a fundamental step toward achieving Artificial General Intelligence (AGI). The central challenge in this paradigm is the "stability-plasticity dilemma": a model must be sufficiently plastic to acquire new knowledge while remaining stable enough to preserve previously learned information. An imbalance in this trade-off, typically skewed toward plasticity, results in "catastrophic forgetting" (French, 1993; Ratcliff, 1990), a phenomenon where performance on prior tasks degrades substantially as new tasks are learned.

To address this challenge, traditional methods like regularization and memory replay have achieved partial success. However, they often introduce significant computational overhead or require access to past data. Recently, a paradigm shift has been catalyzed by the rise of large-scale pre-trained models, particularly Vision Transformers (ViTs) (Dosovitskiy et al., 2021). Their rich, frozen backbones provide inherent resistance to forgetting, pivoting the research focus from training models from scratch towards efficiently adapting these powerful foundations. Within this context, Parameter-Efficient Fine-Tuning (PEFT) (Houlsby et al., 2019; Jia et al., 2022; Zhu et al., 2025), and especially Low-Rank Adaptation (LoRA) (Hu et al., 2022), has emerged as a dominant strategy. For instance, InfLoRA (Liang & Li, 2024), BiLoRA (Zhu et al., 2025) and LoRA⁻DRS (Liu & Chang, 2025) pursue interference-free continual learning by constraining parameter updates to a subspace orthogonal to the knowledge space of previous tasks during training.

The motivation for these orthogonal approaches is to satisfy a fundamental "zero-interference" constraint. That means that to preserve learned knowledge, a parameter update for a new task cannot change the model's output on any historical input. This condition is mathematically guaranteed if the update vector is orthogonal to the subspace containing the features of those inputs. While sound in principle, this orthogonality is inherently violated by the standard LoRA optimization process. Because the optimizer performs independent updates on factors $A$ and $B$, a process which ignores their composite, non-linear effect on the full weight matrix and causes the composite update to deviate from the

required orthogonal path. Furthermore, naively enforcing orthogonality proves counterproductive, severely compromising plasticity. The reason is that it fails to address the deeper, underlying issue of feature-space encroachment: leaving no "free" representational space for new learning. Consequently, the optimizer is placed in a crippling dilemma: any attempt to map new concepts into the occupied feature space is thus heavily penalized by the orthogonality constraint.

To resolve these issues and restore the stability-plasticity balance, we propose Janus-LoRA, a framework that harmonizes parameter-level stability with feature-level plasticity through two novel components. To achieve stability, we first formulate an ideal "safe" gradient direction, $\Delta W_{\text{safe}}$, which lies in a tangent space orthogonal to the task-specific subspace that contains previously learned knowledge. However, directly realizing this direction is non-trivial due to the optimization coupling of LoRA's factors. We address this by deriving Gradient Rectification (GR), a closed-form solution that "reverse-engineers" the ideal $\Delta W_{\text{safe}}$ into decoupled Euclidean updates for factors $A$ and $B$. This rectification mathematically ensures the composite update aligns with the true orthogonal geodesic. Further, to make this practical, GR is integrated with an efficient Online Estimation (OE) algorithm, which tracks the learned task-specific subspace without storing past data.

In addition, to resolve feature space encroachment and enhance plasticity, we introduce Decoupled Margin Loss (DML), which explicitly preserves separation between class spaces. By enforcing a strict angular margin between new data and lightweight historical class prototypes, DML carves out a dedicated, low-interference region for new concepts. This ensures the optimization process for new tasks is not constrained by existing representations, directly enabling the model to learn more effectively, which enhances plasticity and prevents decision ambiguity at task boundaries.

To rigorously validate our framework, we conduct comprehensive experiments on a diverse suite of benchmarks, including ImageNet-R, CIFAR-100, and DomainNet. The results confirm that Janus-LoRA establishes a new state-of-the-art by a significant margin. Its superiority is consistently demonstrated across various datasets, model backbones, and challenging long-sequence scenarios, validating the robustness and general applicability of our principle of balancing parameter-level stability with feature-level plasticity.

Our main contributions are:

- We diagnose catastrophic forgetting in LoRA-based CL as stemming from two core issues: a Parameter-Level Misalignment that violates update orthogonality required for stability, and a subsequent Feature-Space Encroachment that compromises plasticity.

- We propose Janus-LoRA, a framework featuring two novel components: Gradient Rectification (GR) to resolve parameter-level misalignment via closed-form projection, and a Decoupled Margin Loss (DML) to prevent feature-space encroachment.

- Through extensive experiments on diverse benchmarks, we demonstrate that Janus-LoRA establishes a new state-of-the-art, validating that harmonizing parameter-level orthogonality with feature-level separation is a highly effective strategy for continual learning.

## 2. Related Work

### 2.1. Traditional Continual Learning

Continual Learning (CL) aims to enable models to learn from a sequence of tasks without catastrophically forgetting previously acquired knowledge. Traditional CL methods are broadly categorized into three families: regularization-based, expansion-based, and memory-based. **Regularization-based** methods mitigate catastrophic forgetting by penalizing changes to parameters deemed important for previous tasks, either by estimating explicit importance weights (Aljundi et al., 2018; Zenke et al., 2017) or implicitly within a Bayesian framework (Nguyen et al., 2017; Lee et al., 2017). **Expansion-based** approaches, alternatively, dynamically grow the model architecture by adding new sub-networks or modules for each task (Rusu et al., 2016; Veniat et al., 2021). Finally, **memory-based** methods replay data from past tasks, either by storing raw data (experience replay) or generating pseudo-samples (generative replay) (Rebuffi et al., 2017; Sprechmann et al., 2018).

### 2.2. PEFT-based Continual Learning

Parameter-Efficient Fine-Tuning (PEFT), which freezes the backbone and tunes only a small set of parameters, has become the dominant strategy for continual learning with foundation models. One prominent category of PEFT methods is prompt-based tuning. For example, methods like L2P (Wang et al., 2022b), DualPrompt (Wang et al., 2022a), and CODA-Prompt (Smith et al., 2023), learn a pool of prompts and dynamically select relevant ones at inference time. While effective, this selection mechanism adds computational overhead during inference. In contrast, adapter-based methods leverage LoRA (Hu et al., 2022), where a key challenge is preventing updates for a new task from interfering with the adapters of previous tasks. For instance, InfLoRA (Liang & Li, 2024) utilizes gradient projection to design the LoRA matrix $B_t$ such that its update subspace is orthogonal to the gradient spaces of prior tasks. BiLoRA (Zhu et al., 2025) proposes a bilinear framework using fixed bases to achieve "almost-orthogonal" task subspaces probabilistically. Further, LoRA⁻DRS (Liu & Chang, 2025) proposes to first subtract the task vectors of old adapters from the pre-trained

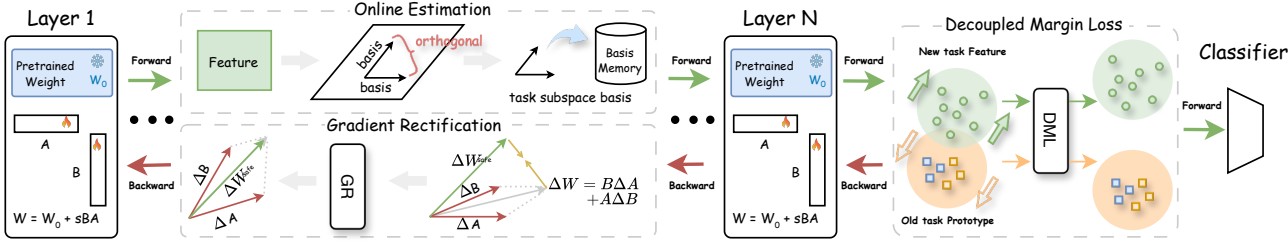

*Figure 1.* Overview of the proposed Janus-LoRA framework.

weights before learning a new task.

While they correctly identify the need for orthogonality between tasks, they attempt to enforce it at a high level without addressing a fundamental flaw in the LoRA optimization process itself. Specifically, independent Euclidean updates to the low-rank factors ($A$ and $B$) cause their composite effect to deviate from any intended orthogonal path. Our work is the first to diagnose this internal misalignment. Gradient Rectification (GR) directly resolves this by mathematically ensuring the final weight update is truly orthogonal, rather than simply projecting the initial gradient.

## 3. Methodology

### 3.1. Preliminaries

**Low-Rank Adaptation (LoRA).** For a pre-trained weight matrix $W_0 \in \mathbb{R}^{d_{out} \times d_{in}}$, LoRA (Hu et al., 2022) freezes $W_0$ and models its update as a low-rank decomposition:

$$W = W_0 + \Delta W = W_0 + sBA, \tag{1}$$

where $A \in \mathbb{R}^{r \times d_{in}}$ and $B \in \mathbb{R}^{d_{out} \times r}$ are the low-rank factors to update, with the rank $r \ll \min(d_{in}, d_{out})$. The scaling factor $s$ is a hyperparameter. This approach dramatically reduces the number of trainable parameters from $d_{in} \times d_{out}$ to $r(d_{in} + d_{out})$, making adaptation highly efficient.

### 3.2. Architecture

Fig. 1 illustrates the overall training pipeline of Janus-LoRA. For each incoming task $\mathcal{D}_t$, the pre-trained backbone $W_0$ is frozen and only the LoRA factors $A$ and $B$ are updated, with the effective weight parameterized as $W = W_0 + sBA$. During training, Online Estimation (OE) uses layer activations to maintain an orthonormal basis $V$ that approximates the protected historical subspace without storing old samples. This basis is used to construct a safe update direction that reduces interference with previously learned knowledge.

At the feature level, Decoupled Margin Loss (DML) encourages compact new-task representations while selectively repelling them from historical class prototypes, preventing feature-space encroachment and preserving plasticity. The task loss and DML are jointly optimized as $L_{\text{total}} =$ $L_{\text{task}} + \lambda L_{\text{DML}}$. During backpropagation, the raw update $\Delta W_{\text{orig}}$ is projected using the estimated historical subspace to obtain the safe update $\Delta W_{\text{safe}} = \Delta W_{\text{orig}}(I - VV^{\top})$. Since LoRA updates the model through factorized parameters rather than the full weight matrix, Gradient Rectification (GR) further converts this safe update into rectified factor updates $\Delta A$ and $\Delta B$. In this way, Janus-LoRA jointly promotes parameter-level stability and feature-level plasticity for exemplar-free continual learning.

### 3.3. Enforcing Orthogonality in Parameter Updates

#### 3.3.1. THE PRINCIPLE: THE IDEAL "SAFE" GRADIENT

The fundamental objective of continual learning is to learn a new task, $\mathcal{T}_t$, without degrading performance on previous tasks, $\mathcal{T}_{1...t-1}$. For a linear layer, this imposes a "zero-interference" constraint: the weight update $\Delta W$ must not alter the output for any historical input activation $a_{\text{past}}$, which implies $\Delta W a_{\text{past}} = 0$. To satisfy this for all past inputs, we aggregate past activations into a matrix $X_{\text{past}}$ and require the update to lie in its null space ($\Delta W X_{\text{past}} = 0$). While satisfying this constraint prevents forgetting, the update must also be effective for the new task. This naturally leads to a constrained optimization problem: finding an update $\Delta W$ that is closest to the original, unconstrained gradient $\Delta W_{\text{orig}} = -\eta \nabla_W \mathcal{L}$ of the new task, while rigorously adhering to the zero-interference constraint:

$$\begin{aligned} \underset{\Delta W}{\text{minimize}} \quad & \frac{1}{2} \|\Delta W - \Delta W_{\text{orig}}\|_F^2 \\ \text{subject to} \quad & \Delta W X_{\text{past}} = 0. \end{aligned} \tag{2}$$

This problem has a closed-form solution, which we formalize in Theorem 3.1.

**Theorem 3.1** (Safe Gradient Projection). *The optimal solution to the constrained optimization problem in Eq. 2, which we term the Safe Gradient ($\Delta W_{safe}$), is given by:*

$$\Delta W_{safe} = \Delta W_{orig} - \Delta W_{orig} P_{past}, \tag{3}$$

*where $P_{past} = X_{past}(X_{past}^T X_{past})^{-1} X_{past}^T$ is the projection matrix onto the historical subspace. (For a detailed derivation, see Appendix A.1).*

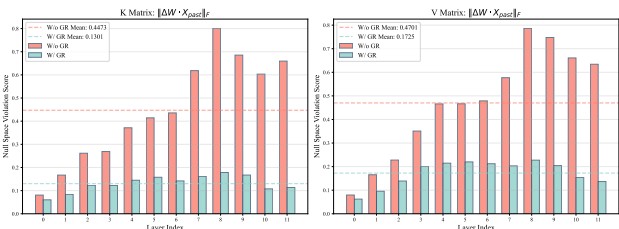

*Figure 2.* **Quantifying Interference from LoRA Updates.** We measure the Null Space Violation ($\|\Delta W \cdot X_{\text{past}}\|_F$) across different model layers. This analysis is performed separately for the LoRA adapters applied to the Key (K) and Value (V) projection matrices within each attention block.

### 3.3.2. GRADIENT RECTIFICATION

The ideal safe gradient, $\Delta W_{\text{safe}}$, provides a theoretical target, but its realization within LoRA's factorized structure is challenging. A naive projection of the factor gradients fails due to a critical issue we term optimization coupling. The gradients of the factors, $\Delta A$ and $\Delta B$, are inherently entangled via the chain rule:

$$\Delta A = (\Delta W_{\text{orig}})^T B \quad \text{and} \quad \Delta B = A^T (\Delta W_{\text{orig}}). \quad (4)$$

An optimizer like Adam (Kingma & Ba, 2015) performs independent Euclidean updates on $A$ and $B$, ignoring the fact that these updates have a coupled and non-linear effect on the full weight matrix $\Delta W_{\text{approx}} = s(B\Delta A + \Delta BA)$ (Wang et al., 2025). Consequently, the composite update does not align with the true "safe" geodesic.

We quantify this failure in Fig. 2. The "Null Space Violation" measures the magnitude of interference ($\|\Delta W X_{past}\|_F$) on previous tasks. As visualized, the standard optimization baseline (red bars) exhibits high violation scores, particularly in deeper layers. This empirical evidence confirms that naive optimization on the factors independently cannot faithfully reconstruct the projected safe gradient $\Delta W_{safe}$.

To resolve this misalignment, we propose Gradient Rectification (GR). Instead of operating on misaligned factor gradients, GR's goal is to "reverse-engineer" the ideal $\Delta W_{\text{safe}}$ into rectified updates, $\Delta A$ and $\Delta B$, such that their composite effect best approximates the target direction. We formulate this as a least-squares problem:

$$\min_{\Delta A, \Delta B} \|\Delta W_{\text{safe}} - s(B\Delta A + \Delta BA)\|_F^2, \quad (5)$$

where $s$ is the LoRA scaling factor.

Simultaneously solving for both $\Delta A$ and $\Delta B$ is a non-convex problem. We therefore adopt a practical and stable hierarchical decomposition approach to find a closed-form solution (as detailed in Appendix A.2). First, we solve for the optimal $\Delta A$ that best approximates $\Delta W_{\text{safe}}$ using the fixed basis $B$. This yields:

$$\Delta A_{\text{rect}} = \frac{1}{s}(B^T B + \delta I)^{-1} B^T \Delta W_{\text{safe}}. \quad (6)$$

Next, we capture the residual error, $R = \Delta W_{\text{safe}} - sB\Delta A_{\text{rect}}$, to solve for the optimal $\Delta B$:

$$\Delta B_{\text{rect}} = \frac{1}{s} R A^T (AA^T + \delta I)^{-1}, \quad (7)$$

where $I$ is the identity matrix and $\delta$ is a small regularization constant for numerical stability.

This hierarchical procedure provides a computationally efficient mechanism to decouple the updates. By explicitly correcting for the coupling effect, the final composite update, formed by $\Delta A_{\text{rect}}$ and $\Delta B_{\text{rect}}$, faithfully aligns with the safe, orthogonal direction dictated by $\Delta W_{\text{safe}}$. This directly prevents parameter-induced forgetting caused by geometric misalignment, as empirically demonstrated in Fig. 2, where our GR (green bars) significantly reduces the null space violation compared to the unrectified baseline (red bars).

### 3.3.3. ENABLING GR WITH ONLINE ESTIMATION

The effectiveness of GR hinges on the projection matrix $X_{\text{past}}$, which defines the historical knowledge subspace to be protected. Conventional approaches for defining the historical subspace, exemplified by the offline procedure in GPM (Saha et al., 2021) and the iterative updates in OWM (Zeng et al., 2019), rely on a suboptimal two-stage process. They necessitate a distinct forward pass performed after task acquisition, a procedure that is not only computationally demanding but also decouples subspace estimation from the primary learning objective.

To overcome this limitation, we propose an Online Estimation (OE) algorithm, which instead formulates basis learning as an integral component of the main optimization. This enables the subspace to be learned concurrently with the task parameters within a single, unified training loop. Specifically, it learns the subspace basis $V \in \mathbb{R}^{D \times k}$, where $k$ denotes the rank of the OE basis $V$, to approximate the principal components of the historical activation space by minimizing a geometric reconstruction error on the current mini-batch of activations $X \in \mathbb{R}^{Batch \times D}$:

$$\mathcal{L}_{\text{recon}}(V) = \frac{1}{2} \|X - XVV^T\|_F^2. \quad (8)$$

The success of this online estimation critically depends on the basis $V$ being orthonormal ($V^T V = I$). This requirement is twofold: it validates $P = VV^T$ as a true projection operator and prevents basis collapse during optimization, where vectors would otherwise become redundant. However, this constraint is incompatible with standard optimizers. A regular Euclidean gradient step, $\tilde{V} \leftarrow V - \eta \nabla_V \mathcal{L}_{\text{recon}}$, will inevitably violate the orthonormality, pushing the basis into an invalid state. Therefore, correctly optimizing $V$ requires moving beyond unconstrained space and treating the optimization as a problem on the set of all orthonormal matrices—a space formally known as the Stiefel manifold (Tunçel, 2009; Bonnabel, 2013).

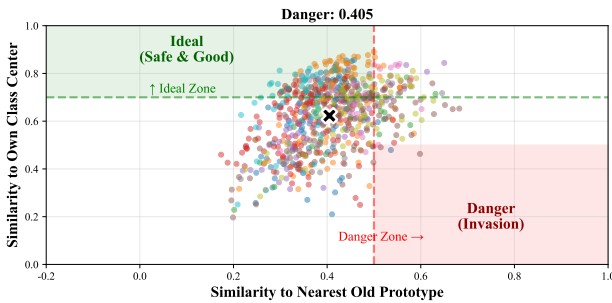

*Figure 3.* **Geometric Effect of DML Learning.** The plots visualize new task features against old and own class prototypes.

To optimize on the Stiefel manifold, we employ Projected Gradient Descent (Rosen, 1960), a standard algorithm for such constrained problems. Specifically, an unconstrained step is taken along the tangent direction:

$$\tilde{V} \leftarrow V - \eta \nabla_V \mathcal{L}_{\text{recon}}(V). \tag{9}$$

Then, this intermediate matrix $\tilde{V}$ is projected back onto the Stiefel manifold via QR decomposition, which efficiently enforces the orthonormality constraint:

$$\text{Let } \tilde{V} = QR \quad \Longrightarrow \quad V \leftarrow Q. \tag{10}$$

This two-step process guarantees that $V$ remains orthonormal after each iteration, enabling OE to supply GR with a dynamic and memory-free approximation of the historical knowledge space while rigorously maintaining the geometric integrity of our framework.

### 3.4. Enforcing Feature-Level Separation

While Sec. 3.3 addresses the *optimization dynamics* by enforcing orthogonality on the parameter level, it does not explicitly constrain the final *geometric structure* of the learned features. This leads the features of a new task to expand uncontrolled, encroaching upon the latent space of old classes—a phenomenon we term "Feature Encroachment". We empirically verify this encroachment by projecting new-task test samples onto a 2D angular plane—with axes representing similarity to the nearest old prototype (X-axis) and the correct new prototype (Y-axis). As shown in Fig. 3, many samples drift into the "Danger Zone" ($\cos \theta_{old} > 0.5$), becoming geometrically closer to old classes than their own.

To resolve this encroachment, our goal is to enforce explicit geometric separation between new data representations and historical class prototypes. We implement this strategy using the Decoupled Margin Loss (DML), a unified objective comprising two key terms. The first term is a contrastive-style (Khosla et al., 2020), prototype-based objective that promotes intra-task compactness by encouraging features of new classes to cluster. The second term is an inter-task hinge penalty that safeguards past knowledge. This term penalizes a new feature $\mathbf{z}_i$ only if its cosine similarity $s_{ij} = \mathbf{z}_i \cdot \mathbf{p}_j$ to

any old prototype $\mathbf{p}_j \in \mathcal{C}_{\text{past}}$ exceeds a predefined margin $m$. The total objective is:

$$\mathcal{L}_{\text{DML}} = - \log \frac{\exp(\mathbf{z}_i \cdot \mathbf{p}_{y_i}/\tau)}{\sum_{c \in \mathcal{C}_t} \exp(\mathbf{z}_i \cdot \mathbf{p}_c/\tau)} + \tag{11}$$
$$\mathbb{E}_{i \in \mathcal{B}, \, \mathbf{p}_j \in \mathcal{C}_{\text{past}}} \left[ \max(0, s_{ij} - m) \right],$$

where $\mathcal{B}$ is the current batch of samples, $\mathbf{p}_{y_i}$ is the prototype for the ground-truth class $y_i$ of sample $\mathbf{z}_i$, $\mathcal{C}_t$ is the set of new classes in the batch, and $\mathcal{C}_{\text{past}}$ is the set of all historical prototypes. The temperature $\tau$ and the margin $m$ are hyperparameters. The intra-task term first establishes well-defined new clusters, which in turn allows the inter-task penalty to apply a precise and minimally invasive repulsive force only when necessary, thus preserving a robust geometric separation between new and old concepts.

The overall training objective of Janus-LoRA combines the standard task-specific objective with our DML into a single, composite loss function. This total loss, $\mathcal{L}_{\text{total}}$, is formulated as a weighted sum:

$$\mathcal{L}_{\text{total}} = \mathcal{L}_{\text{task}} + \lambda_{\text{DML}} \mathcal{L}_{\text{DML}}, \tag{12}$$

where $\mathcal{L}_{\text{task}}$ is the standard cross-entropy loss for the current task, ensuring discriminative learning for new classes. The hyperparameter $\lambda_{\text{DML}}$ balances the contributions of these two objectives. By optimizing this composite loss, Janus-LoRA learns to perform the new task accurately while concurrently preserving the geometric structure of the feature space.

**The Integrated Janus-LoRA Update Process.** The complete process is integrated within each training step. First, a raw gradient, $\nabla_W \mathcal{L}_{\text{orig}}$, is derived from the total loss $\mathcal{L}_{\text{total}}$. Then, this gradient is projected into an ideal "safe" target, $\Delta W_{\text{safe}}$, using an orthonormal basis $V$ provided by our OE algorithm. Finally, GR reverse-engineers this target into rectified updates for the LoRA factors, $A$ and $B$, which are supplied to the optimizer. This integrated mechanism ensures each optimization step works to harmonize parameter-level stability with feature-level plasticity, acting in synergy.

## 4. Experiments

### 4.1. Experimental Settings

**Datasets and Task Setting.** Our experiments are conducted under the Class-Incremental Learning setting, focusing on the challenging Exemplar-Free scenario where the model cannot access data from previous tasks when learning new ones. We evaluate our method on several widely-used continual learning benchmarks: ImageNet-R (Hendrycks et al., 2021): A core evaluation dataset consisting of 200 classes from ImageNet with artistic style renditions. Following prior work, we split it into 5, 10, or 20 sequential tasks. CIFAR-100 (Krizhevsky et al., 2009): A standard CIL benchmark, which we divide into 10 tasks, each con-

taining 10 new classes. DomainNet (Peng et al., 2019): A large-scale dataset spanning multiple visual domains, which we split into 5 tasks. ImageNet-100 (Russakovsky et al., 2015): A subset of the large-scale ImageNet dataset, comprising 100 classes. For our experiments, we divide it into 10 sequential tasks, each introducing 10 new classes.

**Evaluation Metrics.** We follow standard CIL evaluation protocols, using the following metrics to measure performance: **Final Accuracy** ($ACC$): The average accuracy across all classes seen so far, measured after the model has finished training on the final task $T$. $ACC = \frac{1}{T} \sum_{i=1}^{T} A_{T,i}$, where $A_{T,i}$ is the performance on $i$-th task after training the final task ($T$). **Mean Average Accuracy** ($MAA$): This metric computes the average performance across all $T$ tasks, capturing the model's overall performance throughout the entire learning process. $MAA = \frac{1}{T} \sum_{t=1}^{T} \left( \frac{1}{t} \sum_{i=1}^{t} A_{t,i} \right)$. **Backward Transfer** ($BWT$): We use BWT to quantify catastrophic forgetting, where a negative value indicates a performance drop on previous tasks after learning new ones. $BWT = \frac{1}{T-1} \sum_{i=1}^{T-1} (A_{T,i} - A_{i,i})$.

**Baselines.** We compare our proposed method against several state-of-the-art (SOTA) Parameter-Efficient Fine-Tuning (PEFT) based continual learning methods, including: Prompt-based methods: L2P (Wang et al., 2022b), Dual-Prompt (Wang et al., 2022a) and CODA-Prompt (Smith et al., 2023). LoRA-based methods: InfLoRA (Liang & Li, 2024) , LoRA⁻DRS (Liu & Chang, 2025), BiLoRA (Zhu et al., 2025), LoRA-GPM (Saha et al., 2021). Additionally, we include a non-CIL baseline, Finetune, to serve as the performance lower bound. In this setting, the model is sequentially fine-tuned on tasks using standard LoRA without any anti-forgetting mechanism.

**Implementation Details.** Across all benchmarks, we primarily use a Vision Transformer (ViT-B/16) (Dosovitskiy et al., 2021) backbone pre-trained on ImageNet-21K. For all LoRA-based methods, we follow standard practice by inserting adapter modules ($r = 10$) into the Key (K) and Value (V) layers of all attention blocks. To ensure a fair comparison, all LoRA-based baselines in Tab. 2 and Tab. 3 are re-run under a unified implementation protocol based on the InfLoRA codebase, using the same backbone, task splits, optimizer, batch size, training schedule, and random seed settings. Therefore, the numbers reported in our tables are reproduced results under the same controlled setting, rather than results directly copied from the original papers.

Models are trained using the Adam optimizer ($\beta_1 = 0.9, \beta_2 = 0.999$) with a batch size of 128. The number of training epochs is adapted to each dataset (e.g., 50 for ImageNet-R, 20 for CIFAR-100). For our proposed Janus-LoRA, the rank of the OE basis $V$ is set to $k = 50$. In the DML loss, the margin is $m = 0.3$, the loss weight is

*Table 1.* Ablation analysis on the contributions of each module over 10 incremental tasks on ImageNet-R. All reported values are the mean of 5 runs.

| Variant | OE | GR | DML | ImageNet-R | | |
|---|---|---|---|---|---|---|
| | | | | ACC ↑ | MAA ↑ | BWT ↑ |
| ① | | | | $64.24_{0.95}$ | $74.68_{0.98}$ | $-21.85_{0.90}$ |
| ② | | | ✓ | $64.47_{1.07}$ | $75.81_{0.35}$ | $-24.16_{1.33}$ |
| ③ | ✓ | | | $71.82_{0.57}$ | $79.28_{0.49}$ | $-10.13_{0.38}$ |
| ④ | | ✓ | | $70.20_{0.37}$ | $78.80_{0.24}$ | $-14.01_{0.22}$ |
| ⑤ | ✓ | ✓ | | $74.39_{0.73}$ | $80.47_{0.41}$ | $\mathbf{-4.43_{0.60}}$ |
| ⑥ | ✓ | | ✓ | $73.61_{0.05}$ | $80.68_{0.33}$ | $-11.26_{0.11}$ |
| ⑦ | ✓ | ✓ | ✓ | $\mathbf{75.78_{0.16}}$ | $\mathbf{81.64_{0.27}}$ | $-6.05_{0.66}$ |

$\lambda_{\text{DML}} = 1$, and the temperature is set to $\tau = 0.07$, following CLIP (Radford et al., 2021).

### 4.2. Ablation Studies

In this section, we dissect the Janus-LoRA framework through a series of ablation studies. First, we analyze the individual contributions of Online Estimation (OE), Gradient Rectification (GR), and Decoupled Margin Loss (DML) in Tab. 1. We then investigate the impact of the key hyperparameters: the OE basis rank $k$, the DML margin $m$ and loss weight $\lambda_{\text{DML}}$. All studies are conducted on ImageNet-R (10 tasks) and results are averaged over 5 trials.

**Effectiveness of Devised Components.** The results, presented in Tab. 1, break down the performance gains from OE, GR, and DML. As shown in the table, the standard LoRA baseline (①) establishes a lower bound, suffering from severe forgetting. Interestingly, applying DML in isolation (②) boosts new-task accuracy, yet paradoxically exacerbates forgetting. This highlights that simply constraining the feature space is insufficient without first resolving the underlying parameter update misalignment, as the aggressive push for plasticity forces more destructive updates.

Next, we evaluate the components designed to enforce stability through parameter orthogonality. Using only OE to project the gradient (③), significantly reduces forgetting to -10.13. Crucially, its final accuracy of **71.82%** surpasses even the complete LoRA-GPM (71.56% from Tab. 2), powerfully demonstrating the superior effectiveness and data-efficiency of our online estimation strategy. In addition, it is worth noting that the GR-only setting (④) shows a significant improvement. This result shows that merely enforcing gradient-update alignment, a property the baseline lacks, can mitigate forgetting even with a suboptimal target direction. The crucial synergy is revealed in (⑤), where OE and GR are combined. This pairing dramatically reduces forgetting to a mere **-4.43**, the best BWT score in the table. This powerfully confirms our hypothesis: OE

*Table 2.* Class-incremental learning results on the ImageNet-R benchmark across task sequences of varying lengths (T = 5, 10, and 20), comparing our proposed Janus-LoRA with state-of-the-art baselines on a shared ViT-B/16 backbone. The results show that our method significantly outperforms all competitors across all settings, demonstrating superior performance and stability, especially as the number of tasks increases. We report results over 5 trials.

| Model | ImageNet-R (T = 5) | | | ImageNet-R (T = 10) | | | ImageNet-R (T = 20) | | |
|---|---|---|---|---|---|---|---|---|---|
| | ACC ↑ | MAA ↑ | BWT ↑ | ACC ↑ | MAA ↑ | BWT ↑ | ACC ↑ | MAA ↑ | BWT ↑ |
| Fine-Tuning | $69.75_{0.87}$ | $78.93_{0.34}$ | $-17.65_{1.00}$ | $64.24_{0.95}$ | $74.68_{0.98}$ | $-21.85_{0.90}$ | $57.62_{0.87}$ | $69.86_{0.34}$ | $-25.43_{1.11}$ |
| L2P | $68.02_{0.42}$ | $73.68_{0.43}$ | $-7.13_{0.35}$ | $65.99_{0.41}$ | $72.70_{0.53}$ | $-6.95_{0.29}$ | $60.99_{0.20}$ | $68.20_{0.58}$ | $-8.50_{0.68}$ |
| Dual-Prompt | $68.48_{0.19}$ | $72.18_{0.20}$ | $-4.70_{0.40}$ | $64.45_{0.21}$ | $69.23_{0.32}$ | $-4.45_{0.57}$ | $64.45_{0.21}$ | $69.23_{0.32}$ | $-4.45_{0.57}$ |
| Coda-Prompt | $74.05_{0.41}$ | $78.36_{0.17}$ | $-4.61_{0.24}$ | $71.59_{0.66}$ | $76.72_{0.57}$ | $-4.48_{0.47}$ | $65.73_{0.16}$ | $70.98_{0.30}$ | $-4.57_{0.42}$ |
| LoRA⁻DRS | $74.51_{0.66}$ | $80.27_{0.61}$ | $-4.11_{0.49}$ | $72.58_{0.75}$ | $78.34_{0.70}$ | $-3.72_{0.65}$ | $67.60_{0.45}$ | $73.72_{0.43}$ | $\mathbf{-3.66_{0.66}}$ |
| LoRA-GPM | $73.96_{0.72}$ | $81.16_{0.48}$ | $-10.24_{0.79}$ | $71.56_{0.29}$ | $79.34_{0.47}$ | $-10.95_{0.58}$ | $64.06_{0.33}$ | $73.91_{0.45}$ | $-14.56_{0.64}$ |
| BiLoRA | $74.93_{0.28}$ | $78.59_{0.48}$ | $\mathbf{-2.74_{0.15}}$ | $73.93_{0.60}$ | $77.92_{0.52}$ | $\mathbf{-2.07_{0.33}}$ | $61.70_{0.24}$ | $68.60_{0.24}$ | $-5.48_{0.61}$ |
| InfLoRA | $76.31_{0.33}$ | $81.64_{0.34}$ | $-6.17_{0.40}$ | $73.90_{0.69}$ | $80.01_{0.77}$ | $-6.33_{0.56}$ | $68.91_{0.77}$ | $76.24_{0.65}$ | $-7.74_{0.46}$ |
| Janus-LoRA | $\mathbf{77.19_{0.39}}$ | $\mathbf{82.52_{0.35}}$ | $-6.71_{0.50}$ | $\mathbf{75.78_{0.16}}$ | $\mathbf{81.64_{0.27}}$ | $-6.05_{0.66}$ | $\mathbf{71.57_{0.45}}$ | $\mathbf{77.11_{0.35}}$ | $-5.77_{0.46}$ |

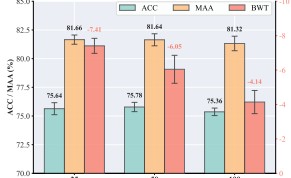

*(a)* Ablation on basis rank $k$.

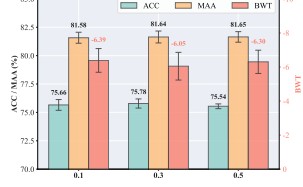

*(b)* Ablation on margin $m$.

*Figure 4.* Ablation study on the rank $k$ and margin $m$ in Janus-LoRA, conducted on ImageNet-R (10 tasks). Error bars represent the standard deviation over 5 independent trials.

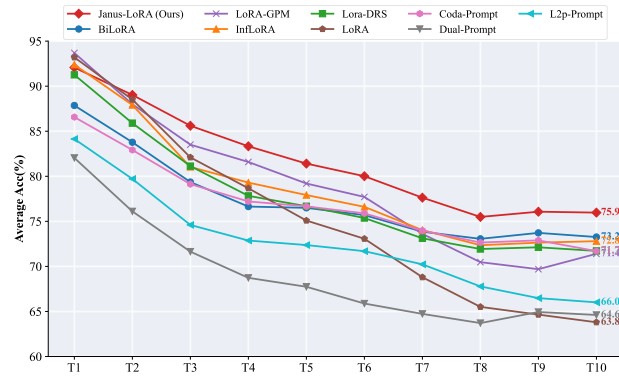

*Figure 5.* Performance Trajectory on ImageNet-R (10 Tasks). The plot shows the average accuracy across all seen tasks, measured after the completion of each new task.

provides the correct subspace to protect, while GR provides the sound mechanism to apply the update. Conversely, combining OE with DML (⑥) achieves a high accuracy but still results in significant forgetting. This shows that the strong, plasticity-driven learning signal from DML cannot be correctly translated into safe parameter updates without GR. Finally, our full Janus-LoRA model (⑦), integrating all components, achieves the highest overall accuracy (ACC and MAA). It reflects a well-managed stability-plasticity trade-off, where the feature separation from DML enhances performance on new tasks at a negligible cost to stability.

**Analysis of Key Hyperparameters.** We analyze three key hyperparameters that govern the stability-plasticity trade-off: the basis rank $k$, the DML margin $m$, and the DML loss weight $\lambda_{DML}$. As shown in Fig. 4 and Fig. 8, our analysis reveals clear trade-offs. For stability, increasing the rank $k$ from 25 to 100 enhances protection of past knowledge (-7.41 to -4.14 in terms of BWT), albeit at a marginal cost to final accuracy. For plasticity, a larger DML margin $m$ or a higher loss weight $\lambda_{DML}$ effectively boosts performance on new tasks (evidenced by rising MAA). However, both exhibit diminishing returns, as excessively high values slightly degrade stability. Based on this analysis, we adopt $k = 50$, $m = 0.3$, and $\lambda_{DML} = 1.0$ as our default settings, as they strike an effective compromise. A more detailed analysis of

each hyperparameter is provided in Appendix B.1.

### 4.3. Experimental Results

Having validated the contributions of each component, we now evaluate the overall performance of Janus-LoRA against SOTA methods on diverse challenging benchmarks.

**Overall Performance.** We begin our comparative analysis on the challenging ImageNet-R benchmark, which is widely used to evaluate robustness against catastrophic forgetting. As shown in Tab. 2, Janus-LoRA consistently establishes a new state-of-the-art across all experimental settings. On longer and more difficult task sequences (e.g., T=10 and T=20), where the risk of forgetting is most severe, the superiority of our method becomes even more pronounced. For instance, in the 20-task setting, Janus-LoRA achieves **77.11%** in terms of MAA, significantly outperforming the next-best competitor, InfLoRA (76.24%). This demonstrates our framework's exceptional stability and its ability to maintain a strong performance trajectory throughout the entire learning process.

*Table 3.* Generalization performance on the **CIFAR-100**, **ImageNet-100**, and **DomainNet** benchmarks. We compare Janus-LoRA against baselines under their respective class-incremental settings to validate the robustness and general applicability of our method. The results demonstrate that our method's superiority is not dataset-specific, as it consistently achieves state-of-the-art performance across these diverse visual domains.

| Model | CIFAR-100 | | | ImageNet-100 | | | DomainNet | | |
|---|---|---|---|---|---|---|---|---|---|
| | ACC ↑ | MAA ↑ | BWT ↑ | ACC ↑ | MAA ↑ | BWT ↑ | ACC ↑ | MAA ↑ | BWT ↑ |
| Fine-Tuning | $83.53_{1.13}$ | $90.18_{0.48}$ | $-11.47_{1.00}$ | $89.10_{0.31}$ | $92.95_{0.33}$ | $-6.80_{0.32}$ | $69.34_{0.09}$ | $74.83_{0.03}$ | $-5.14_{0.20}$ |
| L2P | $84.52_{0.04}$ | $88.74_{0.13}$ | $-4.06_{2.28}$ | $90.66_{0.18}$ | $92.87_{0.29}$ | $-3.02_{0.10}$ | $71.33_{0.14}$ | $76.40_{0.08}$ | $-7.73_{0.56}$ |
| Dual-Prompt | $83.09_{0.49}$ | $88.80_{0.39}$ | $-4.02_{0.32}$ | $89.08_{0.37}$ | $91.79_{0.52}$ | $-3.58_{0.13}$ | $71.70_{0.11}$ | $77.30_{0.07}$ | $-5.74_{0.44}$ |
| Coda-Prompt | $85.36_{0.30}$ | $90.15_{0.16}$ | $-4.41_{0.47}$ | $91.63_{0.38}$ | $93.93_{0.25}$ | $-3.12_{0.23}$ | $73.14_{0.04}$ | $78.64_{0.04}$ | $-5.50_{0.29}$ |
| LoRA⁻DRS | $84.02_{0.85}$ | $89.23_{0.68}$ | $\mathbf{-2.16_{0.33}}$ | $92.42_{0.21}$ | $94.11_{0.11}$ | $\mathbf{-2.03_{0.24}}$ | $73.40_{0.04}$ | $79.36_{0.05}$ | $\mathbf{-4.80_{0.16}}$ |
| LoRA-GPM | $86.01_{0.15}$ | $91.31_{0.14}$ | $-5.56_{0.06}$ | $91.53_{0.59}$ | $94.00_{0.37}$ | $-3.58_{0.30}$ | $69.24_{0.19}$ | $74.84_{0.18}$ | $-5.55_{0.39}$ |
| BiLoRA | $86.48_{0.77}$ | $91.00_{0.44}$ | $-3.32_{0.51}$ | $91.74_{0.29}$ | $93.65_{0.10}$ | $-2.80_{0.25}$ | $67.32_{0.11}$ | $73.03_{0.14}$ | $-4.91_{0.36}$ |
| InfLoRA | $86.69_{0.28}$ | $91.61_{0.23}$ | $-4.61_{0.27}$ | $91.84_{0.32}$ | $93.94_{0.30}$ | $-2.37_{0.05}$ | $72.94_{0.06}$ | $78.98_{0.07}$ | $-7.38_{0.11}$ |
| Janus-LoRA | $\mathbf{88.68_{0.25}}$ | $\mathbf{92.58_{0.20}}$ | $-5.29_{0.33}$ | $\mathbf{92.47_{0.17}}$ | $\mathbf{94.32_{0.14}}$ | $-3.24_{0.46}$ | $\mathbf{73.82_{0.09}}$ | $\mathbf{79.67_{0.05}}$ | $-7.74_{0.17}$ |

Further, Fig. 5 provides a more granular insight into the learning dynamics by visualizing the average accuracy across all seen tasks after each new task is learned. While competitors exhibit a steepening decay, revealing an accumulation of forgetting errors, Janus-LoRA maintains a significantly gentler slope. This demonstrates that our balanced approach to harmonizing stability and plasticity prevents this cumulative decay, leading to a fundamentally more robust continual learner.

**Generalization Across Benchmarks.** To verify the generalization capabilities of Janus-LoRA, we further evaluate it on a diverse suite of benchmarks, including CIFAR-100, ImageNet-100, and the multi-domain DomainNet. The results, summarized in Tab. 3, corroborate our findings from ImageNet-R. Janus-LoRA consistently delivers the highest accuracy across all three datasets, demonstrating clear leadership that spans from standard benchmarks like CIFAR-100—where it surpasses InfLoRA by nearly **2%**—to the large-scale ImageNet-100. More critically, its success on the distributionally-shifted DomainNet is particularly telling. It demonstrates that our framework's ability to resolve geometric conflicts is robust not only to semantic changes but also to drastic shifts in the data domain. This consistent superiority, across benchmarks, complexity, and severe domain variance, provides compelling evidence that our method is a fundamentally robust and widely applicable strategy.

**Robustness to Diverse Pre-training Paradigms.** To demonstrate that the principles of Janus-LoRA are model-agnostic and not contingent on a specific pre-training scheme, we evaluate its performance on ViT backbones trained with diverse paradigms: DINO (Caron et al., 2021) and SAM (Kirillov et al., 2023). The results, summarized in Tab. 4, clearly demonstrate the model-agnosticism of Janus-LoRA. Across both the self-supervised DINO backbone and the segmentation-trained SAM backbone, our method consistently outperforms all baseline methods in terms of

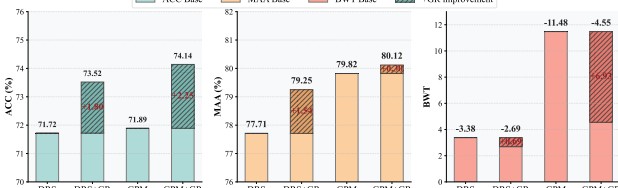

*Figure 6.* Impact of Integrating Gradient Rectification (GR). Performance of LoRA⁻DRS and LoRA-GPM before and after applying the GR module on ImageNet-R (T=10).

both ACC and MAA, establishing a clear lead with 70.22% final accuracy on the latter. This remarkable consistency validates Janus-LoRA as a truly generalizable framework, as it operates on the intrinsic geometric structures inherent to any neural network, regardless of its pre-training paradigm.

**The General Applicability of Gradient Rectification.** To validate our hypothesis that the misalignment between LoRA's actual update and the ideal "safe" gradient is a prevalent issue, we integrated GR as a plug-in module into two distinct methods, LoRA⁻DRS and LoRA-GPM. As shown in Fig. 6, GR yields consistent performance gains across both baselines, confirming the prevalence of the update misalignment issue. This effect is particularly pronounced on LoRA-GPM, where GR boosts ACC by **+2.25%** and BWT by a massive **+6.93**, suggesting it rectifies a particularly severe deviation in its original update process. This experiment thus establishes GR as a **universal, plug-and-play solution** that robustly enhances performance by correcting this fundamental flaw in the LoRA optimization loop.

**Superiority of OE in a Single-Pass Regime.** To simulate the single-pass training regime of large-scale models (Chen et al., 2024), we conducted a direct comparison under a single-epoch constraint. As shown in Tab. 5, OE consistently outperforms GPM, achieving a significant **+3.45%** MAA lead on ImageNet-R. This validates OE as fundamentally more effective and scalable for practical large-scale

*Table 4.* We evaluate all methods on the ImageNet-R (T=10) benchmark using ViT-B/16 backbones pre-trained with DINO and SAM. All experiments are conducted on ImageNet-R (T=10).

| Backbone | Method | ACC ↑ | MAA ↑ | BWT ↑ |
|---|---|---|---|---|
| **SAM** | Fine-Tuning | 47.66 | 55.70 | −40.37 |
| | L2P | 66.01 | 73.21 | −7.75 |
| | Dual-Prompt | 61.19 | 67.77 | −5.27 |
| | Coda-Prompt | 61.92 | 70.88 | −19.01 |
| | LoRA⁻DRS | 68.12 | 75.73 | −4.78 |
| | LoRA-GPM | 58.44 | 66.40 | -26.85 |
| | BiLoRA | 65.66 | 70.95 | **−2.65** |
| | InfLoRA | 69.64 | 77.21 | −5.90 |
| | **Janus-LoRA** | **70.22** | **77.90** | −6.47 |
| **DINO** | Fine-Tuning | 56.79 | 70.50 | −24.36 |
| | L2P | 62.48 | 69.26 | −5.37 |
| | Dual-Prompt | 59.60 | 66.97 | −5.56 |
| | Coda-Prompt | 64.22 | 72.43 | −6.81 |
| | LoRA⁻DRS | 64.39 | 72.07 | −4.36 |
| | LoRA-GPM | 62.15 | 73.34 | -16.56 |
| | BiLoRA | 65.84 | 72.33 | **−3.91** |
| | InfLoRA | 67.83 | 75.98 | −7.79 |
| | **Janus-LoRA** | **68.20** | **76.15** | −5.73 |

*Table 5.* Performance comparison between OE and GPM under a single-epoch training regime on ImageNet-R and CIFAR-100.

| Dataset | Method | ACC ↑ | MAA ↑ | BWT ↑ |
|---|---|---|---|---|
| **ImageNet-R** | GPM | $38.79_{0.42}$ | $42.48_{0.22}$ | $-5.36_{0.55}$ |
| | OE | $\mathbf{40.90_{0.47}}$ | $\mathbf{45.93_{1.15}}$ | $-5.98_{0.90}$ |
| **CIFAR100** | GPM | $74.84_{1.85}$ | $83.30_{1.78}$ | $-7.70_{0.32}$ |
| | OE | $\mathbf{77.31_{0.63}}$ | $\mathbf{84.83_{0.80}}$ | $\mathbf{-6.27_{2.06}}$ |

continual learning, as its incremental basis construction is inherently more data-efficient than GPM.

**Dissecting the Geometric Mechanism of DML.** To provide direct empirical validation for the mechanism of DML, we now analyze its geometric impact on the feature space. The result is visualized in Fig. 7a. The DML loss effectively evacuates the danger zone, compressing the feature distribution towards the safe subspace ($\cos\theta_{old} < 0.2$). Additionally, it can be observed that DML explicitly trades some intra-class compactness for more robust geometric separation, a visually confirmed trade-off driven by the final gradient's balance between the intra-task "pull" and the inter-task "push". In addition, in Fig. 7b, we plot the maximum similarity of new samples to old prototypes, where high values indicate feature overlap. While the baseline (Red) exhibits a heavy-tailed distribution of high-risk similarities, Janus-LoRA (Blue) drastically suppresses the probability density in this region. This provides quantitative validation that our DML successfully mitigates feature-space encroachment by enforcing a margin against historical prototypes.

**Efficiency Analysis.** To assess practical efficiency, we benchmarked the total end-to-end training time and instantaneous throughput on the 10-task ImageNet-R sequence. The results, detailed in Fig. 9, 10, 11 in the Appendix, reveal that Janus-LoRA is the most efficient method overall, completing the entire benchmark in just 1197.8 seconds. This superior performance stems directly from our fully online architectural design, which eliminates the costly, data-dependent offline processing required by methods like GPM. While our online modules introduce a modest computational cost per step, the time saved by avoiding the offline stage yields a substantial net gain in overall training speed. Crucially, this efficiency is achieved without impacting deployment, as all methods share identical inference FLOPs. A more detailed analysis is provided in Appendix B.2.

## 5. Conclusion

In this work, we revisit catastrophic forgetting in LoRA-based continual learning from a unified geometric perspective. Rather than treating parameter-level update misalignment and feature-space encroachment as two isolated problems, we view them as two coupled aspects of the stability-plasticity conflict under LoRA adaptation. To address this conflict, we propose Janus-LoRA, a unified framework named after Janus, the two-faced god in Roman mythology, to reflect its dual focus on coordinating parameter-level stability and feature-level plasticity. Gradient Rectification (GR) improves stability by making LoRA updates better follow the intended safe direction, while Decoupled Margin Loss (DML) mitigates feature-space encroachment and preserves new-task learning capacity. Across challenging benchmarks, Janus-LoRA achieves state-of-the-art performance, showing that jointly addressing these coupled geometric effects is an effective and principled strategy for LoRA-based continual learning.

## Acknowledgements

This study is supported by grants from the Fundamental and Interdisciplinary Disciplines Breakthrough Plan of the Ministry of Education of China (No. JYB2025XDXM116), the National Natural Science Foundation of China (Grant No.62425208, No. U22A2097, No. U23A20315, No. 62402094) and Fundamental Research Funds for the Central Universities.

## Impact Statement

This paper presents work whose goal is to advance the field of machine learning. There are many potential societal consequences of our work, none of which we feel must be specifically highlighted here.

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

# A. Proof

## A.1. Proof of Safe Gradient Projection

In this section, we provide the detailed mathematical derivation for Theorem 3.1 presented in the methodology. We demonstrate that the optimal update $\Delta W_{safe}$, which minimizes the Euclidean distance to the original gradient while satisfying the zero-interference constraint, is obtained via orthogonal projection onto the null space of the reference activations.

Let $\Delta W \in \mathbb{R}^{d_{out} \times d_{in}}$ be the weight update matrix we seek to optimize. Let $\Delta W_{orig} \in \mathbb{R}^{d_{out} \times d_{in}}$ be the unconstrained gradient update computed from the current task loss (i.e., $\Delta W_{orig} = -\eta \nabla_W \mathcal{L}_{task}$). Let $X_{\text{past}} \in \mathbb{R}^{d_{in} \times N}$ be the matrix of input activations from previous tasks, where $N$ is the number of reference samples.

The constrained optimization problem is formally defined as:

$$\underset{\Delta W}{\text{minimize}} \quad J(\Delta W) = \frac{1}{2} \|\Delta W - \Delta W_{orig}\|_F^2, \quad (13)$$

$$\text{subject to} \quad \Delta W X_{\text{past}} = \mathbf{0}, \quad (14)$$

where $\| \cdot \|_F$ denotes the Frobenius norm. To solve this problem, we employ the Method of Lagrange Multipliers. The Lagrangian function $\mathcal{L}(\Delta W, \Lambda)$ is formulated as:

$$\mathcal{L}(\Delta W, \Lambda) = \frac{1}{2} \|\Delta W - \Delta W_{orig}\|_F^2 + \text{Tr}\left(\Lambda^T (\Delta W X_{\text{past}})\right), \quad (15)$$

where $\Lambda \in \mathbb{R}^{d_{out} \times N}$ is the matrix of Lagrange multipliers corresponding to the equality constraint $\Delta W X_{\text{past}} = \mathbf{0}$. We use the trace operator $\text{Tr}(\cdot)$ to express the inner product between matrices in the constraint term.

Using the identity $\|\mathbf{A}\|_F^2 = \text{Tr}(\mathbf{A}^T \mathbf{A})$, we expand the objective term:

$$\begin{aligned} \frac{1}{2} \|\Delta W - \Delta W_{orig}\|_F^2 = \\ \frac{1}{2} \text{Tr}\left((\Delta W - \Delta W_{orig})^T (\Delta W - \Delta W_{orig})\right). \end{aligned} \quad (16)$$

We verify the first-order optimality condition by taking the partial derivative of the Lagrangian with respect to $\Delta W$ and setting it to zero:

$$\frac{\partial \mathcal{L}}{\partial \Delta W} = (\Delta W - \Delta W_{orig}) + \Lambda X_{\text{past}}^T = \mathbf{0} \quad (17)$$

$$\Rightarrow \quad \Delta W = \Delta W_{orig} - \Lambda X_{\text{past}}^T. \quad (18)$$

To determine the value of $\Lambda$, we substitute Eq. (18) back into the primal constraint $\Delta W X_{\text{past}} = \mathbf{0}$:

$$(\Delta W_{orig} - \Lambda X_{\text{past}}^T) X_{\text{past}} = \mathbf{0}, \quad (19)$$

$$\Delta W_{orig} X_{\text{past}} - \Lambda (X_{\text{past}}^T X_{\text{past}}) = \mathbf{0}. \quad (20)$$

Assuming the correlation matrix $X_{\text{past}}^T X_{\text{past}}$ is invertible (or utilizing the Moore-Penrose pseudoinverse if singular), we

solve for $\Lambda$:

$$\Lambda = \Delta W_{orig} X_{\text{past}} (X_{\text{past}}^T X_{\text{past}})^{-1}, \qquad (21)$$

Finally, we substitute the expression for $\Lambda$ back into Eq. (18) to obtain the closed-form solution for the optimal update $\Delta W$:

$$\Delta W = \Delta W_{orig} - \left(\Delta W_{orig} X_{\text{past}} (X_{\text{past}}^T X_{\text{past}})^{-1}\right) X_{\text{past}}^T \qquad (22)$$

$$= \Delta W_{orig} - \Delta W_{orig} \left(X_{\text{past}} (X_{\text{past}}^T X_{\text{past}})^{-1} X_{\text{past}}^T\right) \qquad (23)$$

$$= \Delta W_{orig} \left(I - X_{\text{past}} (X_{\text{past}}^T X_{\text{past}})^{-1} X_{\text{past}}^T\right). \qquad (24)$$

Let $\mathcal{P}_{past} = X_{\text{past}} (X_{\text{past}}^T X_{\text{past}})^{-1} X_{\text{past}}^T$. This matrix represents the orthogonal projection operator onto the column space of $X_{\text{past}}$ (the feature subspace of previous tasks). Consequently, the optimal update is:

$$\Delta W_{safe} = \Delta W_{orig}(I - \mathcal{P}_{past}). \qquad (25)$$

This confirms that $\Delta W_{safe}$ corresponds to the projection of the original gradient onto the null space of $X_{\text{past}}$ (represented by the projection operator $I - \mathcal{P}_{past}$), thereby concluding the proof of Theorem 3.1.

### A.2. Derivation of Gradient Rectification

In this section, we derive the closed-form solutions for the *Gradient Rectification* introduced in Section 3.3.2. Our goal is to find the optimal LoRA parameter updates $\Delta A$ and $\Delta B$ such that their combined effect best reconstructs the theoretically safe gradient $\Delta W_{safe}$.

Recall that the LoRA weight update is factorized as $\Delta W_{approx} = s(B\Delta A + \Delta BA)$, where $s$ is the scaling factor. We aim to minimize the reconstruction error with respect to the safe gradient target $\Delta W_{safe}$:

$$\underset{\Delta A, \Delta B}{\text{minimize}} \quad \mathcal{J} = \frac{1}{2} \|\Delta W_{safe} - s(B\Delta A + \Delta BA)\|_F^2. \qquad (26)$$

Solving for $\Delta A$ and $\Delta B$ simultaneously is a non-convex problem due to the product interaction (if we were solving for $A$ and $B$ from scratch) or highly coupled (in the update case). To ensure a stable and unique solution, we adopt a hierarchical decomposition approach: First, optimize $\Delta A$ to capture the projection of $\Delta W_{safe}$ onto the current column space of $B$. Second, optimize $\Delta B$ to capture the residual error that lies in the orthogonal complement of $B$.

We assume $\Delta B = 0$ for this step and focus on finding the optimal coefficients $\Delta A$ that best approximate $\Delta W_{safe}$ given the fixed basis $B$. The objective function reduces to:

$$\mathcal{J}_A = \frac{1}{2}\|\Delta W_{safe} - sB\Delta A\|_F^2 + \frac{\gamma}{2}\|\Delta A\|_F^2. \qquad (27)$$

Here, we introduce a Tikhonov regularization term (with coefficient $\gamma$) to ensure numerical stability, preventing coef-

ficient explosion when $B$ is ill-conditioned.

Taking the derivative with respect to $\Delta A$:

$$\frac{\partial \mathcal{J}_A}{\partial \Delta A} = -sB^T(\Delta W_{safe} - sB\Delta A) + \gamma \Delta A. \qquad (28)$$

Setting the derivative to zero:

$$s^2 B^T B \Delta A + \gamma \Delta A = sB^T \Delta W_{safe}, \qquad (29)$$

$$(s^2 B^T B + \gamma I)\Delta A = sB^T \Delta W_{safe}. \qquad (30)$$

Solving for $\Delta A$:

$$\Delta A = s(s^2 B^T B + \gamma I)^{-1} B^T \Delta W_{safe}. \qquad (31)$$

To simplify implementation and align with standard ridge regression notation, let $\delta = \gamma/s^2$. The update becomes:

$$\Delta A = \frac{1}{s}(B^T B + \delta I)^{-1} B^T \Delta W_{safe}. \qquad (32)$$

This solution effectively preconditions the update by the inverse covariance of $B$, rectifying distortions caused by the singular value distribution of $B$.

After determining $\Delta A$, we compute the residual component of the safe gradient that could not be represented by the fixed basis $B$:

$$R = \Delta W_{safe} - sB\Delta A. \qquad (33)$$

Substituting Eq. 32 (with $\delta \to 0$) into the residual, we get $R \approx (I - B(B^T B)^{-1} B^T)\Delta W_{safe}$. This shows that $R$ corresponds to the projection of $\Delta W_{safe}$ onto the *orthogonal complement* of the column space of $B$. Thus, the columns of $R$ are orthogonal to $B$.

We now solve for $\Delta B$ to approximate this residual. The update equation is $s\Delta BA \approx R$. The objective function is:

$$\mathcal{J}_B = \frac{1}{2}\|R - s\Delta BA\|_F^2 + \frac{\gamma}{2}\|\Delta B\|_F^2. \qquad (34)$$

Taking the derivative with respect to $\Delta B$:

$$\frac{\partial \mathcal{J}_B}{\partial \Delta B} = -s(R - s\Delta BA)A^T + \gamma \Delta B, \qquad (35)$$

Setting to zero:

$$s^2 \Delta BAA^T + \gamma \Delta B = sRA^T, \qquad (36)$$

$$\Delta B(s^2 AA^T + \gamma I) = sRA^T. \qquad (37)$$

Solving for $\Delta B$:

$$\Delta B = sRA^T(s^2 AA^T + \gamma I)^{-1}. \qquad (38)$$

Similarly, defining $\delta = \gamma/s^2$, we obtain the final update rule:

$$\Delta B = \frac{1}{s}RA^T(AA^T + \delta I)^{-1}. \qquad (39)$$

This step aligns the rotation of the subspace (represented by $\Delta B$) with the residual error $R$, normalized by the activation covariance of $A$. Since $R$ lies in the null space of $B^T$, this update ensures that $\Delta B$ primarily rotates the subspace into new directions required by the safe gradient, decoupling the rotation dynamics from the scaling dynamics handled by $\Delta A$.

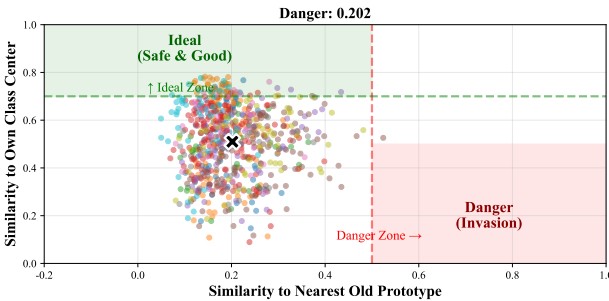

*(a)* Geometric Effect of DML Learning.

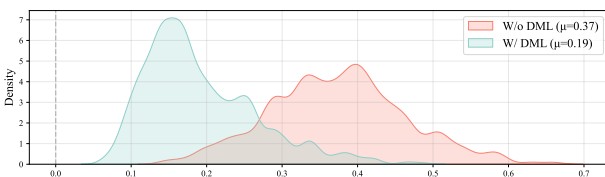

*(b)* Density distribution of the maximum cosine similarity between current task samples and historical prototypes.

*Figure 7.* Visual and Statistical Analysis of DML.

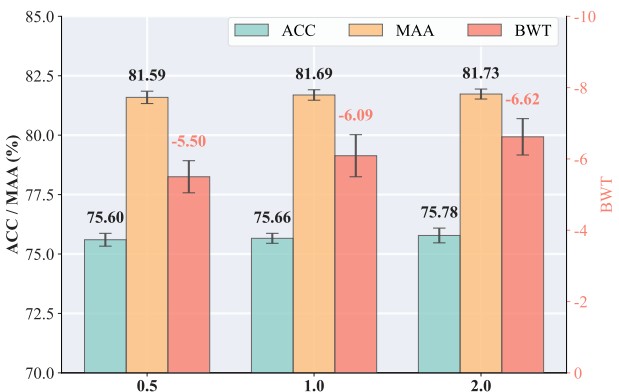

*Figure 8.* Ablation study on the DML loss weight $\lambda_{\text{DML}}$ in Janus-LoRA, conducted on ImageNet-R (10 tasks). Error bars represent the standard deviation over 5 independent trials.

## B. Results

### B.1. Ablation Results Analysis

**Effect of Basis Rank** $k$**.** We investigate the impact of the historical subspace rank $k$, which determines the dimensionality of the subspace basis protected by OE. As shown in Fig. 4a, a clear trade-off between stability and plasticity emerges. As $k$ increases from 25 to 100, BWT improves significantly (from -7.41 to -4.14), confirming that a larger basis provides a more complete representation of past knowledge, thus offering stronger protection against forgetting. Conversely, this stronger constraint marginally reduces plasticity, reflected in a slight decrease in final accuracy. Based on our analysis showing that final performance is robust to the rank $k$, we adopt $k = 50$ as it provides an effective

trade-off between stability and plasticity.

**Effect of margin** $m$**.** We then analyze the DML margin $m$, which controls the repulsive force separating new and old feature spaces. As shown in Fig. 4b, increasing the margin $m$ effectively boosts plasticity, evidenced by the rising MAA. However, we observe a subtle side effect where a very large margin ($m = 0.5$) begins to negatively impact stability, slightly degrading the BWT score. This suggests a point of diminishing returns where the benefit to plasticity is outweighed by the harm to stability. Consequently, $m = 0.3$ is chosen as it strikes the best compromise, offering strong plasticity without a noticeable penalty to stability.

**Effect of DML loss weight** $\lambda_{\text{DML}}$**.** We analyze the impact of the DML loss weight, $\lambda_{\text{DML}}$, which balances the contribution of the feature separation objective against the primary task loss. As shown in Fig. 8, increasing $\lambda_{\text{DML}}$ from 0.5 to 2.0 progressively enhances plasticity, reflected in the steady rise of both final accuracy (ACC) and mean average accuracy (MAA). This indicates that a stronger push for feature-level separation effectively improves the model's ability to learn new tasks. However, this gain in plasticity comes at the cost of stability. The Backward Transfer (BWT) score degrades from -5.50 to -6.62, suggesting that an overly aggressive feature separation objective can slightly interfere with the preservation of past knowledge. Consequently, we select $\lambda_{\text{DML}} = 1.0$ as it offers a robust balance, achieving strong performance on new tasks without a significant penalty to stability.

### B.2. Further Efficiency Analysis.

We benchmarked the total training time and throughput on the 10-task ImageNet-R sequence (5 epochs per task). To ensure robustness, results are reported as the mean and standard deviation over 5 complete runs with different random seeds.

Fig. 9 plots the training time for each of the 10 individual tasks, where a clear pattern emerges: Janus-LoRA (red line) is consistently the most efficient method. For instance, on the final task, it requires only 141.4 seconds, while competitors like InfLoRA take over 160 seconds. This superior per-task efficiency stems from our architectural design. While much of our time savings comes from eliminating the costly offline processing of methods like GPM, this graph specifically proves our online components are also exceptionally lightweight.

The similar fluctuation patterns across all methods are due to the varying number of classes and samples in each task of the ImageNet-R benchmark. However, Janus-LoRA's consistent lead within these fluctuations proves its efficiency is robust and not task-dependent. This sustained per-task speed advantage is the primary driver of its overall end-to-

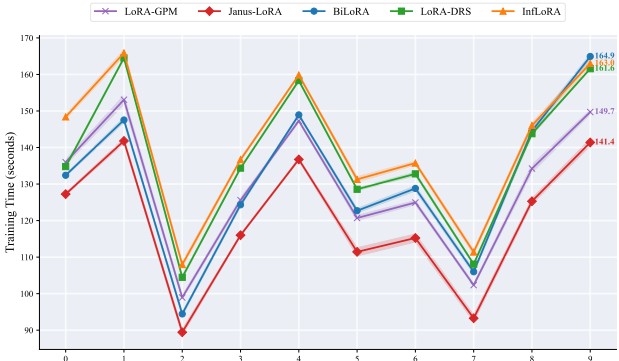

*Figure 9.* Per-Task Training Time. This plot shows the wall-clock time required to complete 5 epochs for each of the 10 individual tasks in the ImageNet-R sequence, averaged over 5 random seeds. The results highlight that Janus-LoRA is consistently the most efficient method within each task.

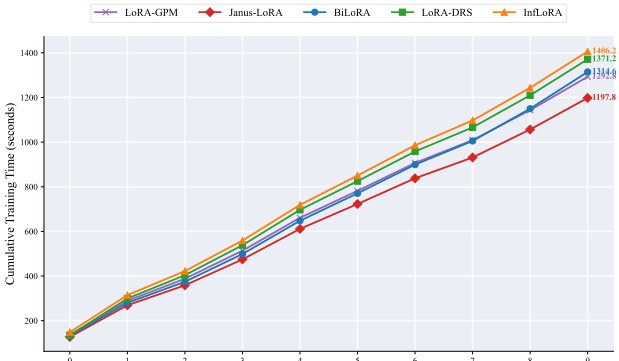

*Figure 10.* Cumulative End-to-End Training Time. This figure tracks the cumulative training time across the entire 10-task ImageNet-R sequence. Janus-LoRA demonstrates superior overall efficiency, completing the benchmark fastest.

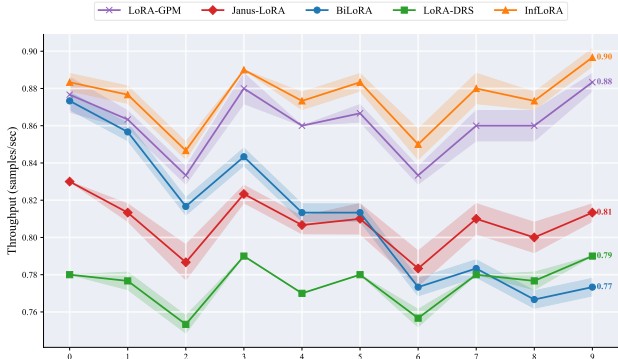

*Figure 11.* Instantaneous Throughput. This graph illustrates the instantaneous throughput (samples per second) during the training process. The throughput of Janus-LoRA is only marginally lower than the Lora-GPM and InfLoRA; this slight reduction is an expected and direct consequence of the per-step computations performed by our OE and its GR mechanism.

Fig. 10.

A key aspect of our work is that training acceleration does not come at the cost of inference overhead. We confirmed that all evaluated methods, including Janus-LoRA, produce a final model with identical deployment costs: 33.726 GFLOPs and 85.662M parameters. This proves that our proposed training modules (OE and GR) are a training-only construct that leaves no computational footprint on the final model. Therefore, the significant training speed-up offered by Janus-LoRA is achieved at zero cost to inference efficiency, making it a practically advantageous solution for real-world continual learning.

end efficiency.

The definitive end-to-end efficiency of Janus-LoRA is shown in Fig. 10. Our method completes the 10-task sequence in only 1197.8 seconds, substantially outpacing all competitors. This widening gap is a direct result of our fully online design, which circumvents the burdensome offline processing of methods like GPM, causing its initial speed advantage to compound with each new task.

Finally, Fig. 11 provides a granular view of online computational cost by plotting instantaneous throughput. The graph reveals that Janus-LoRA's throughput is slightly lower than some baselines, which is the intentional and minimal overhead introduced by our OE and its GR mechanism to enforce orthogonality in each step. This result critically validates our design as a strategic trade-off: the minor per-step cost of maintaining orthogonality is vastly outweighed by the elimination of severe offline processing bottlenecks, ultimately leading to the superior overall efficiency demonstrated in

