# OpenReview forum: "JANUS-LORA: A Balanced Low-Rank Adaptation for Continual Learning"
_ICML.cc/2026/Conference — ICML 2026 regular_

### Official Review · Reviewer_VpiL · 2026-02-19

**Soundness:** 4
**Presentation:** 4
**Significance:** 3
**Originality:** 3
**Overall Recommendation:** 5
**Confidence:** 3

**Summary:**

This paper proposes a new efficient mechanism for class incremental learning involving LoRAs. It first identifies a weakness in existing approaches involving adapters, namely that updates to adapters, if if their original gradient is projected, can lead to inference with learned tasks. Ideally, a zero-inference regime is desired, meaning no forgetting. To approach this, the authors, second, then provide a mechanisms that rectifies the gradients of the LoRA submatrices. Because this mechanism depends on identifying occupied subspaces, the authors, third, use an online estimation to build subspace matrices. Finally, representational (dis-) similarity is enforced by using a decoupled margin loss that separates the feature space representation of data samples. Extensive experiments on large-scale CL datasets demonstrate the effects of the proposed components.

**Compliance With Llm Reviewing Policy:**

Affirmed.

**Final Justification:**

Explanantion:

My initial recommendation was "5-Accept". The rebuttal has not changed this recommendation in any way. Concretely, for me the reasons for acceptance are (i) proposing gradient mechanisms for the common LoRA setting, (ii) extensive experiments on **large** CL datasets, and (iii) the presentation of the paper.

During the rebuttal, very minor unclarities have been resolved.

**Key Questions For Authors:**

I only have minor points that I provide below.

(1) The name "Janus" is not explained. While it might be known for european readers, adding a short paragraph in the appendix about its dual nature might help.

(2) Are the numbers for Fig. 4 for the task-specific test sets?

(3) Regarding Table 1, experiment 3: which gradient is projected if only OE is used? I undestood that OE is a basis for GR, so if ONLY OE is used, what happens? Does it only affect learning V (which, in turn, affects feature representations)?

(4) Are the used models frozen and only the new adapters trained? Or is the entire backbone (+ adapaters) trained?

**Limitations:**

yes

**Strengths And Weaknesses:**

- The presentation. I found the paper easy to follow, noticing a red thread throughout. Also, the general layout (figures, tables, etc.) is clean.
- Well-supported claims: The paper includes a throughout ablation study (Table1) that clearly analyses the claimed effects from the components (grad. rectification, OE, DML)
- Good combination and improvement of related ideas in CL. (1) While the idea of projecting gradients is not new (authors themselves acknowledge GPM), existing works require a two-pass to build projection bases. Authors improve this to learn bases online during training. (2) The idea of prototypes has been explored by previous works (eg. [1]), but used here together with a margin component.
- Extensive experiments help understand the method. Tables 1, 2, 3, 4 and accompanying figures provide detailed results on common, challenging CL datasets. The results all help to increase the understanding of the proposed method, Janus-LoRA, and its workings.

[1] Sarfraz, Fahad, Elahe Arani, and Bahram Zonooz. "Semantic aware representation learning for lifelong learning." The Thirteenth International Conference on Learning Representations. 2025.

---

> ### Author Rebuttal · Authors · 2026-03-31
>
> ```
> 1. The name "Janus" is not explained. While it might be known for european readers, adding a short paragraph in the appendix about its dual nature might help.
> ```
> Thank you for the suggestion. In the revised manuscript, we will briefly explain the meaning of “Janus” in the appendix. The name comes from Janus, the two-faced god in Roman mythology, and reflects that our method addresses two coupled but competing aspects of continual learning: stability and plasticity.
>
> ```
> 2. Are the numbers for Fig. 4 for the task-specific test sets?
> ```
> Thank you for pointing this out. The numbers in Fig. 4 are the average accuracy over all seen tasks/classes after training on each task. They are not task-specific test-set accuracies. We will make this clearer in both the figure caption and the main text.
>
> ```
> 3. Regarding Table 1, experiment 3: which gradient is projected if only OE is used? I undestood that OE is a basis for GR, so if ONLY OE is used, what happens? Does it only affect learning V (which, in turn, affects feature representations)?
> ```
> Thank you for pointing out this ambiguity.
>
> **First**, in the “OE only” setting, the projected quantity is the **current LoRA update direction**, i.e., the update applied to the LoRA factors \(A\) and \(B\). The role of OE here is to estimate the historical subspace basis online.
>
> **Second**, if only OE is used, GR is not applied. **The update is simply projected** according to the subspace estimated by OE, without correcting the mismatch caused by LoRA’s factorized updates.
>
> **Third**, yes, in the “OE only” setting, **OE directly learns and maintains the basis \(V\)**. However, its role does not stop there. Once learned, \(V\) is used to define the protected subspace and project the current LoRA update. Therefore, **OE affects feature representations indirectly**. We will revise the manuscript to make this setting more explicit.
>
> ```
> 4. Are the used models frozen and only the new adapters trained? Or is the entire backbone (+ adapaters) trained?
> ```
> Yes. We follow the standard PEFT/LoRA setting: the **pretrained backbone is frozen**, and **only the newly introduced LoRA/adapters are trained**.
> We will state this more explicitly in the implementation details section.

---

> > ### Author Rebuttal · Reviewer_VpiL · 2026-04-01
> >
> > Thank you for poviding a rebuttal to my review. I acknowledge the authors effort and trust them to integrate the changes (Janus name, OE gradient setting) into the paper.
> >
> > My question (2) was likely not precisely enough written, so I have a follow-up: is the data used to compute the average accuracy over all tasks from the test or the training data?
> >
> > --- Edit: authors replied to my question below (thanks!). My concerns are resolved. I will maintain my rating of "5 - Accept" ---

---

> > > ### Author Response · Authors · 2026-04-01
> > >
> > > ```
> > > Is the data used to compute the average accuracy over all tasks from the test or the training data?
> > > ```
> > >
> > > Thank you for the follow-up. We apologize for the ambiguity. The values in Fig. 4 are computed on the **test sets**, not on the training data.
> > >
> > > More specifically, each point at task \(T_t\) is obtained **after finishing training on task \(t\)**. At that point, we evaluate the current model on the **test sets of all tasks seen so far** (i.e., tasks \(1,$\dots$,t\)), and average these test accuracies to obtain the plotted value. Therefore, the point at \(T_{10}\) is the average test accuracy after training has been completed on all 10 tasks. We will clarify this explicitly in the figure caption and the main text.

---

### Official Review · Reviewer_16Cg · 2026-03-05

**Soundness:** 2
**Presentation:** 3
**Significance:** 2
**Originality:** 2
**Overall Recommendation:** 3
**Confidence:** 4

**Summary:**

The paper present a Low-Rank (LoRA) approach for continual learning. The key idea is to enforce an orthogonality constraint on the LoRA updates with respect to the learned subspace via a projected gradient descent to address the updates does not to unlearning the earlier task. Further, to address the plasticity, i.e., the ability to learn new tasks, the paper proposed an approach to decouple the margin loss to promote feature separation.

**Compliance With Llm Reviewing Policy:**

Affirmed.

**Final Justification:**

Based on the theoretical advancement or experimental results of this paper, I consider this a weak reject paper.

**Key Questions For Authors:**

1.  It is unclear why orthoganality is an ideal/preferred approach to achieve stability of the model? Why is it guaranteed that the new task features lie in an orthogonal space when compared to earlier task. It may not hold in related yet different task setting.

2. How does the problem in Equation (2) enforce the low-rank constraint? Theorem 1 does not seems to be a formulation of the problem the paper is trying to solve as it ignores the low-rank constraint on the model.

3. The motivation for the hierarchical solution in Equations (6) and (7) is not understood. Standard approaches include alternating optimization techniques, including Altmin, AltProj-GD and so on and the paper claims practical and stable approach without clear detail on why these are efficient. Finally, in section 3.2.3 the paper uses a Proj-GD approach, however, why this method is chosen is unclear.

4. What is the key novelty in the feature separation technique proposed in the paper as compared to existing methods?

**Limitations:**

The techniques or metrics presented in the paper does not seems to novel to my understanding and hence I do not recommend this paper for publication in ICML. The specific comments and questions are given above.

**Strengths And Weaknesses:**

1. The paper is easy to read and addresses the stability-plasticiy trade-off in LoRA models.

2. It is unclear why orthoganality is an ideal/preferred approach to achieve stability of the model? Why is it guaranteed that the new task features lie in an orthogonal space when compared to earlier task. It may not hold in related yet different task setting.

3. The approach proposed to address stability, in my opinion, is not very novel given that it used Projected Gradient Descent (Proj-GD) for ensuring the orthogonality, which is a long-time known technique in constrained optimization. Similarly, for feature separation, using the cosine similarity based metrics has been studied.

4. The presentation in the paper is rather confusing at times, where the paper present problems that do not fully capture the original problem that is being studied and then presents discussion on why the solution obtained has some challenges and then proposes another and so on.   The paper introduces multiple solution strategies (subsections 3.2.1, 3.2.2, and 3.2.3), but these methods seem to address derivative issues introduced by earlier limitations rather than the problem originally stated. As a result, the connection between the problem formulation and the proposed solutions is unclear. Some of the parameters like $\Delta A$ and $\Delta B$ dimensions are not defined, explicitly. In line 191, B is reused to denote a matrix dimension. What is k in line 188?

---

> ### Author Rebuttal · Authors · 2026-03-31
>
> ```
> 1. Why is orthogonality a preferred choice for stability? This may not hold in related-but-different task settings.
> ```
> Thank you for the question.
> **For the first part**, orthogonality is used because our derivation shows that if the update lies in the null space of the historical subspace, then it satisfies the zero-interference condition (Theorem 3.1). In this sense, orthogonality provides a principled target for preserving old knowledge. Prior orthogonality-based continual-learning methods also suggest that this can be effective in practice as a stability mechanism.
>
> **For the second part**, we do **not** assume that new-task features are orthogonal to old-task features. Instead, orthogonality is imposed on the **update**. We use it as a conservative constraint to protect old knowledge, even in related-but-different task settings. Task relatedness may affect how much useful update remains after projection, but the projected update still lies in the orthogonal complement of the historical subspace.
>
> ```
> 2. Eq. (2) and Theorem 3.1 seem to ignore the low-rank LoRA constraint. How do they relate to the actual problem you solve?
> ```
> Thank you for pointing this out. **Eq. (2) does not enforce the low-rank LoRA constraint**. Instead, it defines **an ideal target** that we want the update to satisfy during learning, whether the model is parameterized in full-rank form or under LoRA.
>
> Similarly, Theorem 3.1 is not the problem we finally solve under LoRA. Instead, it provides a **sufficient condition for stability** in continual learning, namely the ideal safe gradient. **The core problem of our paper is to resolve the gap** between this safe gradient and what can actually be realized under LoRA’s factorized update rule.
> We will revise the manuscript to make this distinction between the ideal target and its LoRA-constrained realization more explicit.
>
> ```
> 3. The motivation for the hierarchical solution in Eqs. (6) and (7), and for using Proj-GD in Section 3.2.3, is unclear.
> ```
> Thank you for the question.
>
> For Eqs. (6) and (7), the two factor updates (ΔA, ΔB) are **coupled** under LoRA parameterization. Joint optimization would require solving a coupled subproblem at every step. Iterative alternatives such as alternating optimization are possible, but they **add inner-loop updates and increase complexity**. We therefore adopt the hierarchical solution: first solve for ΔA with fixed B, then solve for ΔB on the residual. This **closed-form decomposition** is lighter and easier to integrate into standard LoRA training.
>
> For Section 3.2.3, Proj-GD is used to preserve the **orthogonality of the subspace basis**. This is needed to define a valid projection operator and avoid basis collapse. A standard Euclidean update would break this constraint, while Proj-GD keeps the basis feasible after each step.
>
> ```
> 4. The stability part seems based on standard projected optimization, and the feature-separation part seems based on known cosine-similarity ideas. What is the novelty?
> ```
> Thank you for this comment. We agree that our method is related to projected-gradient approaches. However, our contribution is not projection itself, but a **LoRA-specific failure mode** that standard projection methods do not resolve: under LoRA factorization, independently updating A and B does not guarantee that the composed full-matrix update follows the intended safe direction. In this sense, our novelty lies in a **LoRA-compatible realization** of the projection objective.
>
> For feature separation, DML is novel because it is both decoupled and selective. It separates intra-task compactness from inter-task separation, and its repulsion term is activated **only** when a new feature enters the neighborhood of a historical prototype. This directly targets **feature encroachment** while avoiding unnecessary constraints on new-task learning. Since DML is also replay-free and prototype-based, it is well suited to exemplar-free LoRA continual learning.
>
> ```
> 5. The connection between the problem formulation and Sections 3.2.1 / 3.2.2 / 3.2.3 is unclear. Some symbols and dimensions are also undefined.
> ```
> Thank you for pointing this out. **The core problem here is a single one**: how to realize zero interference in LoRA-based continual learning so that old knowledge can be preserved. Sections 3.2.2 and 3.2.3 do not introduce new goals; they address the **two realization challenges** of the same objective.
>
> Section 3.2.1 defines the ideal target. Section 3.2.2 addresses the LoRA realization gap, where independent updates to A and B deviate from the ideal direction. Section 3.2.3 addresses the online realization gap, since the protected historical subspace must be maintained during training. Together, they **make the ideal target realizable** under LoRA parameterization and training constraints.
> We will revise the manuscript to make this logic clearer and to define the missing symbols and dimensions more explicitly.

---

> > ### Author Rebuttal · Reviewer_16Cg · 2026-04-04
> >
> > Thank you for addressing my comments. I have no further questions, and I have increased my score.

---

### Official Review · Reviewer_9KbK · 2026-03-17

**Soundness:** 3
**Presentation:** 4
**Significance:** 3
**Originality:** 3
**Overall Recommendation:** 4
**Confidence:** 4

**Summary:**

This work proposes a principled approach to mitigating interference in LoRA updates. The authors show that enforcing orthogonality in the individual LoRA matrices A and B is insufficient, as interference can still arise from their combined effect. To address this issue, the paper derives a mathematically grounded procedure for computing matrix updates that explicitly accounts for the interaction between the two matrices. This procedure is supported by an online estimation mechanism that leverages historical information while remaining memory-free. In addition, the authors introduce a new loss formulation designed to push the representations of newly introduced classes away from previously learned ones, reducing representation overlap during incremental learning.

**Compliance With Llm Reviewing Policy:**

Affirmed.

**Final Justification:**

The authors provided a clear and constructive rebuttal that addressed several of the concerns raised in the initial review, which I appreciate. However, while the responses were helpful and clarified certain aspects, they do not sufficiently change my overall assessment of the paper’s soundness, originality, and significance, and therefore my initial score remains unchanged.

**Key Questions For Authors:**

The proposed method strongly emphasizes enforcing orthogonality in the LoRA updates. However, in continual learning settings, strict orthogonality constraints may limit positive knowledge transfer between tasks. Have the authors investigated whether this constraint negatively affects beneficial transfer across tasks?

**Limitations:**

yes

**Strengths And Weaknesses:**

Strengths

* The paper clearly describes the main components of the proposed method, particularly Gradient Rectification and the Decoupled Margin Loss, providing sufficient technical detail for understanding the approach.

* The experimental setup is well documented, including datasets, evaluation protocols, and metrics, which supports reproducibility.

* The authors present comprehensive ablation studies that carefully analyze the contribution of each component of the method.

* The experimental evaluation is extensive and shows that the proposed approach generally achieves higher performance than competing baselines.

Weaknesses

* The paper would benefit from a visual overview of the method, such as a diagram summarizing the overall workflow and the interaction between the different components.

* In Table 2, the reported results for InfLoRA are noticeably lower than those reported in the original paper. When comparing with the original results, the proposed method would outperform the baseline only marginally in two scenarios and underperform in one. A similar discrepancy appears for other baselines such as LoRA-DRS and BiLoRA. The authors should clarify the cause of this discrepancy, as it raises questions about the experimental setup and the relative effectiveness of the proposed approach.

* The evaluation does not include comparisons with other recent LoRA-based approaches such as SD-LoRA [1]. According to the results reported in the SD-LoRA paper, its performance appears to be higher than the results presented here. Including this method in the comparison would provide a clearer picture of the relative performance of the proposed approach.

[1] Wu, Yichen, et al. "SD-LoRA: Scalable Decoupled Low-Rank Adaptation for Class Incremental Learning." The Thirteenth International Conference on Learning Representations.

---

> ### Author Rebuttal · Authors · 2026-03-31
>
> ```
> 1. The paper would benefit from a visual overview of the method, such as a diagram summarizing the overall workflow and the interaction between the different components.
> ```
> Thank you for the suggestion. We agree that an overview figure would improve readability. In the revised manuscript, we will add a method diagram to show the roles of OE, GR, and DML, and how they interact within one training step.
>
> ```
> 2. In Table 2, the reported results for InfLoRA are noticeably lower than those reported in the original paper. When comparing with the original results, the proposed method would outperform the baseline only marginally in two scenarios and underperform in one. A similar discrepancy appears for other baselines such as LoRA-DRS and BiLoRA. The authors should clarify the cause of this discrepancy, as it raises questions about the experimental setup and the relative effectiveness of the proposed approach.
> ```
> Thank you for raising this important point. We agree that this discrepancy should be clarified. The main reason is that the numbers in Table 2 are not copied from the original papers. Instead, **we re-ran all baselines under a unified implementation protocol** based on the InfLoRA codebase, with the same backbone, task splits, training schedule, LoRA rank, optimizer, batch size, and random seed settings. As a result, our reported numbers reflect **performance under one controlled setting** rather than each method’s original environment.
> We will make this explicit in the revised manuscript and clearly state that all results in Table 2 are our own reproduced results under one unified setting.
>
> ```
> 3. The evaluation does not include comparisons with other recent LoRA-based approaches such as SD-LoRA. According to the results reported in the SD-LoRA paper, its performance appears to be higher than the results presented here. Including this method in the comparison would provide a clearer picture of the relative performance of the proposed approach.
> ```
> Thank you for the suggestion. Following your comment, we reproduced SD-LoRA under our experimental pipeline on the **ImageNet-R** benchmark. Since SD-LoRA and InfLoRA differ in whether task order is shuffled, we report two settings for completeness: the no-shuffle protocol used in our main table, and the shuffled protocol used in the original SD-LoRA implementation.
>
> Under the unified no-shuffle protocol, Janus-LoRA consistently outperforms SD-LoRA in both the 5-task and 10-task settings. On 5-task, Janus-LoRA improves ACC by `+1.94` and MAA by `+2.35`, while keeping comparable BWT. On the more challenging 10-task setting, the gain is larger: Janus-LoRA improves ACC by `+6.71`, MAA by `+2.63`, and BWT by `+7.11`, showing clearly better performance and much less forgetting.
>
> For completeness, we also report results under the shuffled protocol used by SD-LoRA. In that setting, Janus-LoRA still achieves higher ACC and MAA in both the 5-task and 10-task settings. We will add these results and clarify the protocol difference in the revised manuscript.
>
>
> |Method||5| | |10| |
> |------------------|-----|-----|------|-----|-----|------|
> ||ACC|MAA|BWT|ACC|MAA|BWT|
> |Finetune|69.75|78.93|−17.65|64.24|74.68|−21.85|
> |SD-LoRA|75.25|80.17|-6.73|69.07|79.01|-13.16|
> |Janus-LoRA|77.19|82.52|−6.71|75.78|81.64|−6.05|
> |SD-LoRA-Shuffle|78.10|82.05|-6.65|75.92|81.86|-5.57|
> |Janus-LoRA-Shuffle|79.23|83.25|-5.34|77.13|82.74|-6.77|
>
> ```
> 4. The proposed method strongly emphasizes enforcing orthogonality in the LoRA updates. However, in continual learning settings, strict orthogonality constraints may limit positive knowledge transfer between tasks. Have the authors investigated whether this constraint negatively affects beneficial transfer across tasks?
> ```
> Thank you for this important question. **Yes, we have investigated this issue in our ablation study**. The `OE + GR` variant enforces parameter-level orthogonality most strongly. As shown in Table 1, it achieves the best BWT (`-4.43`), which means the strongest forgetting suppression, but its ACC / MAA are only `74.39 / 80.47`, not the best overall. This suggests that strong orthogonality alone can protect old tasks, but may also **limit beneficial transfer** and reduce new-task plasticity.
>
> After adding DML, the full Janus-LoRA improves ACC / MAA to `75.78 / 81.64`, while BWT only slightly drops to `-6.05`. This shows that **DML helps recover the plasticity** reduced by strict orthogonality while keeping strong stability. Therefore, our goal is not to enforce orthogonality as strongly as possible, but to **achieve a better balance between stability and plasticity**.
> We will add this analysis more explicitly to the main text in the revised manuscript.

---

> > ### Author Rebuttal · Reviewer_9KbK · 2026-04-02
> >
> > Thank you for addressing my comments. I have no further questions and I will keep my score.

---

### Official Review · Reviewer_zK5p · 2026-03-19

**Soundness:** 3
**Presentation:** 3
**Significance:** 3
**Originality:** 2
**Overall Recommendation:** 4
**Confidence:** 3

**Summary:**

This paper addresses catastrophic forgetting in continual learning with pre-trained ViTs using LoRA adapters.
The authors identify two distinct sources of forgetting: (1) parameter-level misalignment where naive LoRA updates violate orthogonality to previous tasks, and (2) feature-level encroachment where new task representations drift into the space of old classes.
To address these, they propose Janus-LoRA with two key components: Gradient Rectification (GR) , a closed-form method to align LoRA factor updates with an ideal "safe" gradient direction orthogonal to past task subspaces, and Decoupled Margin Loss (DML) , which combines intra-task contrastive learning with an inter-task margin penalty against old class prototypes.
An Online Estimation (OE) algorithm tracks the historical subspace without storing past data.
Experiments on ImageNet-R, CIFAR-100, and DomainNet show the outperformance, with ablations aim at validating each component.

**Compliance With Llm Reviewing Policy:**

Affirmed.

**Key Questions For Authors:**

Aside from those above, could you clarify the theoretical and empirical computational efficiency improvements claimed in the paper? I see your arguments in the Appendix but it is not completely clear to me on this subject.

**Limitations:**

No negative societal impact

**Strengths And Weaknesses:**

I think this paper is, in its current form, a borderline paper that I lean toward acceptance.
The paper makes a compelling argument that forgetting in LoRA-based continual learning stems from two distinct geometric issues (parameter orthogonality violation and feature-space encroachment).
I have some hesitancy about the extent of novelty of the proposed method and the soundness of the experiment, which I hope the authors can provide some clarification.

- Soundness: I believe the algorithm design is mathematically sound enough for an empirical paper. What I am doubting is that in the experiment section, for the Transformer-based architecture, the authors only provide experiments on ViT with vision tasks. Since there is no implication that the method is not applicable to the language field where Transformer and LoRA are vital, some proof-of-concept experiments in the language field are highly desired.

- Presentation is good. The paper is easy to read and to follow.

- Significance: I think the works address a relevant problem in the CL + PEFT field. It would be significant enough in the CL + Large Model field with language experiments.

- Originality: I hope for some clarification here.
  - There are some PEFT-based CL methods and the author mentioned some. Could you please clarify how your method is different? In particular, it seems to me that the safe gradient projection resembles GPM.
  - Also, the paper states that "we identify that catastrophic forgetting in LoRA-based continual learning stems from two distinct geometric issues", where both issues have been studied separately. I suggest the authors could temper the claim and position the contribution as a unified framework.

Overall, I think this is a valid work that I would hope for some clarification on its relevance and contribution to the CL + PEFT field.

---

> ### Author Rebuttal · Authors · 2026-03-31
>
> ```
> 1. The paper only evaluates ViT on vision tasks. A proof-of-concept experiment in the language domain would be valuable.
> ```
> Thank you for the suggestion. We agree that a language-side proof-of-concept would strengthen the paper. Our method is not vision-specific; it addresses a LoRA-specific optimization issue under factorized updates.
>
> Following your suggestion, we conducted experiments on the CLIP text branch using the official ZSCL codebase [1] and the X-TAIL benchmark [2]. Due to the limited rebuttal period, we were only able to complete the experiments reported below. Janus-LoRA achieves the **best performance** under the 10-task setting, providing initial evidence that our method also extends to language-side LoRA adaptation.
>
>
> |Method||10| |
> |----------|-------------|------------|---------|
> ||Transfer Mean|Average Mean|Last Mean|
> |Finetune|61.6|56.6|44.2|
> |LoRA-DRS|62.1|66.3|71.1|
> |LoRA-GPM|61.8|66.9|73.4|
> |Janus-LoRA|**62.7**|**68.0**|**75.2**|
>
> [1] Zheng, Zangwei, et al. "Preventing zero-shot transfer degradation in continual learning of vision-language models." ICCV, 2023.
> [2] Xu, Yicheng, et al. "Advancing cross-domain discriminability in continual learning of vision-language models." NeurIPS, 2024.
>
> ```
> 2. Please clarify how your method differs from PEFT-based CL methods, especially GPM-like approaches.
> ```
> Thank you for the comment. Unlike prior PEFT-based methods such as InfLoRA, BiLoRA, and LoRA-DRS, which mainly impose constraints at the update or subspace level, our work identifies a **LoRA-specific optimization issue**: under factorized LoRA updates, independently updating A and B does not guarantee that the composed full-matrix update follows the intended safe direction.
>
> This is also the main difference from GPM. GPM mainly addresses where to project the update, while our method addresses how to **realize that projected safe direction** under LoRA factorization. This is why we introduce GR, which reconstructs the ideal safe full-matrix update in factor space through a closed-form rectification.
>
> ```
> 3. Also, the paper states that "we identify that catastrophic forgetting in LoRA-based continual learning stems from two distinct geometric issues", where both issues have been studied separately. I suggest the authors could temper the claim and position the contribution as a unified framework.
> ```
> Thank you for this suggestion. We agree that our original wording can be improved. More precisely, our contribution is not to claim two separate discoveries, but **a unified view of forgetting** in LoRA-based continual learning. The two phenomena in the paper are not isolated problems. **They are two coupled aspects of the same stability-plasticity conflict**.
>
> Specifically, parameter-level update misalignment explains why the intended interference-free update may **fail to preserve stability** under LoRA factorization. Feature-space encroachment explains why stability alone can **reduce plasticity** for new tasks. Our goal is therefore not to present two unrelated contributions, but to show that **these two effects must be addressed together**. Following your suggestion, we will revise the wording and present Janus-LoRA more clearly as a unified framework that coordinates parameter-level stability and feature-level plasticity.
>
> ```
> 4. Please clarify the claimed theoretical and empirical efficiency improvements.
> ```
> Thank you for pointing this out.
>
> **From a theoretical perspective**, Janus-LoRA and GPM-like projection methods share the same main online safe-projection term $O(LSH^2k)$, where $H$ is the hidden dimension, $k$ is the historical subspace rank, $L$ is the number of LoRA insertion layers, and $S=En/b$ is the number of training steps. The difference comes from the extra overhead.
> For GPM-like methods, the extra cost is the task-end offline update:
> $$
> \Delta C_{\mathrm{proj}} = O(nC_{\mathrm{fwd}}) + O(LnH^2).
> $$
> For Janus-LoRA, the extra cost comes from online OE and GR updates:
> $$
> \Delta C_{\mathrm{Janus}} = O\left(LEnHk + \frac{LEnHk^2}{b} + \frac{LEnH^2r}{b}\right),
> $$
> where $r$ is the LoRA rank.
> Under our setting $H=768$, $k=50$, $r=10$, $b=128$, $E=50$, $L=24$, this gives:
> $$
> \Delta C_{\mathrm{proj}} \approx O(nC_{\mathrm{fwd}}) + O(1.51\times 10^7 n), \qquad
> \Delta C_{\mathrm{Janus}} \approx O(1.19\times 10^8 n).
> $$
> So Janus-LoRA is faster whenever:
> $$
> C_{\mathrm{fwd}} > 1.04\times 10^8.
> $$
> Using the reported ViT-B/16 forward FLOPs (33.726 GFLOPs) as a coarse order-of-magnitude proxy for $C_{\mathrm{fwd}}$, this condition is clearly satisfied in practice.
>
> **Empirically**, this matches our appendix results. Janus-LoRA has slightly lower instantaneous throughput because OE and GR add per-step computation, but it achieves better per-task and cumulative end-to-end training time because it removes the extra task-end subspace update stage.
> This gain comes with no extra deployment cost, since the final inference FLOPs and parameter count remain unchanged.

---

> > ### Author Rebuttal · Reviewer_zK5p · 2026-04-04
> >
> > Thank you for addressing my comments. I have no further questions and I will keep my score.

---

### Decision · Program_Chairs · 2026-04-30

**Decision:**

Accept (regular)

**Comment:**

This paper studies continual learning with LoRA adapters and argues that forgetting arises from a mismatch between the intended safe update direction and the factorized update actually realized by LoRA, together with feature-space interference that can hurt plasticity. To address this, it proposes Janus-LoRA, combining Gradient Rectification, Online Estimation, and a Decoupled Margin Loss. The reviewers generally found the paper technically solid, clearly written, and supported by strong experiments and ablations, with overall scores of 5/4/4/3. The main concerns were novelty relative to prior projection-based continual learning methods, limited evaluation breadth beyond vision, discrepancies with originally reported baseline numbers, and some presentation clarity. In the rebuttal, the authors clarified the LoRA-specific distinction from GPM-like approaches, added initial language-side results, explained that baseline numbers were reproduced under a unified protocol, provided additional comparisons with SD-LoRA, and addressed questions about orthogonality, transfer, and notation; these responses were viewed positively by the reviewers, with concerns largely resolved and no score decreases. Overall, I find that the paper makes a meaningful and sufficiently supported contribution to continual learning with parameter-efficient adaptation, and I recommend Accept.